# Dermal PapillaCell-Derived Exosomes Regulate Hair Follicle Stem Cell Proliferation via LEF1

**DOI:** 10.3390/ijms24043961

**Published:** 2023-02-16

**Authors:** Jiali Li, Bohao Zhao, Shuyu Yao, Yingying Dai, Xiyu Zhang, Naisu Yang, Zhiyuan Bao, Jiawei Cai, Yang Chen, Xinsheng Wu

**Affiliations:** 1College of Animal Science and Technology, Yangzhou University, 48 South University Ave Yangzhou, Yangzhou 225009, China; 2Joint International Research Laboratory of Agriculture & Agri-Product Safety, Yangzhou University, 48 South University Ave Yangzhou, Yangzhou 225009, China

**Keywords:** exosomes, hair follicle stem cell, dermal papilla cells, hair follicle, LEF1

## Abstract

Hair follicle (HF) growth and development are controlled by various cell types, including hair follicle stem cells (HFSCs) and dermal papilla cells (DPCs). Exosomes are nanostructures that participate in many biological processes. Accumulating evidence indicates that DPC-derived exosomes (DPC-Exos) mediate HFSC proliferation and differentiation during the cyclical growth of hair follicles. In this study, we found that DPC-Exos increase ki67 expression and CCK8 cell viability readouts in HFSCs but reduce annexin staining of apoptotic cells. RNA sequencing of DPC-Exos-treated HFSCs identified 3702 significantly differentially expressed genes (DEGs), including *BMP4*, *LEF1*, *IGF1R*, *TGFβ3*, *TGFα*, and *KRT17*. These DEGs were enriched in HF growth- and development-related pathways. We further verified the function of LEF1 and showed that overexpression of LEF1 increased the expression of HF development-related genes and proteins, enhanced HFSC proliferation, and reduced HFSC apoptosis, while knockdown of LEF1 reversed these effects. DPC-Exos could also rescue the siRNA-LEF1 effect in HFSCs. In conclusion, this study demonstrates that DPC-Exos mediated cell-to-cell communication can regulate HFSCs proliferation by stimulating LEF1 and provide novel insights into HF growth and development regulatory mechanisms.

## 1. Introduction

Both animal and human hair have functions in defense, protection, communication, and aesthetics [1,2]. Hair follicles (HFs) function as small independent organs and have attracted much attention in studies of embryonic development [3,4]. Because of their high self-renewal ability and distinct structural characteristics, HFs are good scientific models for addressing critical biological questions such as cell proliferation, differentiation, and aging [5]. Hair follicles continuously undergo anagen, catagen, and telogen during cyclic growth [6,7,8,9]. The cellular components of HFs and the localization of hair follicle cells are well studied; the periodic growth of the hair cycle is also well described [10,11]. Two types of cells play vital roles in HF growth, including hair follicle stem cells (HFSCs) at the prominent bulge region and dermal papilla cells (DPCs) at the bottom of the HF [5,12]. Molecule signals from HFSCs are crucial for guiding the arrangement of various epidermal elements around the hair follicle [13], while the number of DPCs defines the size, shape, and cycling frequency of hair cells, and reduction in DPCs causes follicular decline [14].

Extracellular vesicles with a diameter of 30–150 nm, also known as exosomes, are critical carriers of intercellular communication and play essential biological functions by transferring nanoscale molecules, such as mRNAs, miRNAs, and proteins, between cells [15]. In addition to conveying messages between cells, exosomes can also remodel the extracellular matrix [16]. There is abundant evidence demonstrating exosome functions in various systems, including immunity, pregnancy, embryonic development, tissue repair, cardiovascular calcification and bone remodeling, liver homeostasis, and nervous system regulation [17,18,19,20,21,22,23,24,25]. In HFs, DPC-derived exomes (DPC-Exos) have been shown to regulate hair follicle growth and development [26]. DPC-Exo treatment increases outer root sheath cell migration and proliferation and regulates HF development via β-catenin and Shh pathways [27,28].

Hair follicle morphogenesis is regulated by many signaling pathways, including Wnt, hedgehog, bone morphogenetic protein (BMP), transforming growth factor beta (TGF-β), and Notch [29,30,31,32,33,34,35]. LEF1 is necessary for the transduction of Wnt signaling and is crucial for the development of HF. Highly expressed during the anagen stages of the hair cycle [36], LEF1 can regulate the differentiation of bulge stem cells by promoting the translocation of β-catenin into the nucleus [37].

Yan H et al. used miRNA sequencing between DPCs and DPC-Exos and found that the miR-22-5p–LEF1 axis regulates HFSC proliferation [28]. However, the roles of DPC-Exos in regulating HFSC proliferation remain unclear. This study found that DPC-Exos can directly regulate HFSC proliferation and apoptosis. Then, a transcriptomic analysis between HFSCs with and without DPC-Exos was made. HFSCs treated with DPC-Exos showed significant changes in 3702 differentially expressed genes, including 1904 upregulated and 1798 downregulated genes. Among these genes, LEF1 was significantly increased in HFSCs cocultured with DPC-Exos. Further investigations showed that LEF1 could promote HFSC proliferation and inhibit HFSC apoptosis. Thus, DPC-Exos likely regulate HF growth and development by stimulating LEF1 in HFSCs. These findings provide novel insights into exosome-mediated cell-to-cell communication regulating HF growth and development.

## 2. Results

### 2.1. DPC-Exos Could Be Uptake into HFSCs

DPC-Exos were first characterized at the structural and protein levels by transmission electron microscopy (TEM), nanoparticle tracking analysis (NTA), and protein level detection, respectively. The ultrastructure of DPC-Exos showed the double-layered membrane typical of exosomes (Figure 1A). NTA showed particle sizes of 79.49 nm ± 18.92 nm (Figure 1B). DPC-Exos expressed CD9 and TSG101. However, calnexin was not detected, indicating that DPC-Exos were not contaminated with cellular debris (Figure 1C). Adding DiI-labeled DPC-Exos (DiI-DPC-Exos) to HFSC cultures and incubating for 24 h resulted in red fluorescence in the HFSCs, confirming that the DPC-Exos can enter HFSCs and mediate intercellular communication (Figure 1D).

### 2.2. DPC-Exos Promote HFSC Proliferation and Inhibit HFSC Apoptosis

In order to explore the functions of DPC-Exos in HFSCs, HFSCs were incubated with DPC-Exos for 48 h. The HFSC-control group was increased by 100 µL PBS for negative control. DPC-Exos-treated HFSCs (HFSCs + Exos) showed significantly increased ki67 staining (Figure 2A). CCK8 cell viability assay confirmed that DPC-Exos promote HFSC proliferation from 48 h to 72 h (Figure 2B), while annexin staining showed decreased apoptosis in HFSCs treated with DPC-Exos (Figure 2C).

### 2.3. DPC-Exos Induce Differential Gene Expression in HFSCs

HFSCs with and without DPC-Exos treatment were used for RNA sequencing to determine how DPC-Exos affects HFSCs at the gene level. Differentially expressed genes from the two treatment groups were analyzed with the Pheatmap R package. Samples were clustered according to the expression levels of the same gene across different samples and the expression patterns of different genes within the same sample. Hierarchical clustering was performed using the longest distance method (Complete Linkage), where the distance was calculated using the Euclidean formula (Figure 3A). A total of 16,452 genes were identified in the HFSC + Exos and HFSC-control groups. Of these, 3702 DEGs (1904 upregulated and 1798 downregulated genes, Appendix A) were selected based on log2-fold-change ≥1 and *p* < 0.05, including *LAMC1*, *BMP4*, *LEF1*, *FGF5*, *Wnt10b*, *FOSL1*, and *KRT17* (Figure 3B). To verify the sequencing data, a random selection of DEGs were verified by qPCR. Figure 3C shows that *BMP4*, *LEF1*, *IGF1R,* and *TGFB3* were significantly increased in DPC-Exos-treated HFSCs, while *TGFα*, *FGFR1*, *FOSL1*, and *KRT17* were significantly decreased. GO functional enrichment and KEGG pathway analyses were carried out to reveal the potential biological function of these DEGs. The enriched GO terms include extracellular matrix organization, multicellular organismal process, cell surface reporter signing pathway, and developmental process. In particular, numerous DEGs were enriched for GO terms associated with hair follicle growth and development, such as animal organ development (Figure 3D). In the KEGG pathway analysis, hair growth- and development-related pathways such as ECM-receptor interaction, Wnt, and hedgehog signaling pathways were also enriched (Figure 3E). Table 1 summarizes the HF growth- and development-related DEGs. These results suggest that DPC-Exos may regulate the function of HFSCs and affect hair growth.

### 2.4. LEF1 Regulates the Expression of HF Growth- and Development-Related Genes and Proteins

The RNA-seq data above showed an approximately 1.3-fold increase in *LEF1* expression in DPC-Exos-treated HFSCs, and previous studies revealed that *LEF1* promotes HF growth and regulates HF cells [28,38,39,40]. Thus, we hypothesized that DPC-Exos might promote HF growth by stimulating *LEF1* expression in HFSCs. HFSCs were transfected with the expression vector pcDNA3.1-LEF1, which elevated LEF1 in HFSCs and increased the expression of HF growth-related genes such as *BCL2*, *BMP2*, *CCND1*, *CTNNB1*, and *EGF* (Figure 4A). In contrast, siRNA targeting of LEF1, particularly siRNA-1915, decreased LEF1 expression in HFSCs and downregulated *BCL2*, *BMP2*, *CCND1*, *CTNNB1*, and *EGF* expression (Figure 4B). We further confirmed at the protein level that the CCND1 level is dependent on LEF1 expression (Figure 4C).

### 2.5. LEF1 Enhances Proliferation and Inhibits Apoptosis in HFSCs

LEF1 has been shown to have an antiapoptotic role [40]. Here, it has been shown that overexpression of LEF1 (pcDNA3.1-LEF1) significantly promoted HFSC proliferation from 24 to 72 h, while knockdown of LEF1 with siRNA decreased the viability of HFSCs compared to siRNA-NC (Figure 5A). Conversely, overexpression of LEF1 inhibited apoptosis, while knockdown of LEF1 enhanced HFSCs apoptosis (Figure 5B), indicating that LEF1 has a positive effect on HFSC proliferation.

### 2.6. DPC-Exos Could Rescue siRNA-LEF1 Effect in HFSCs

To further research the rescue effect of DPC-Exos on the knockdown of LEF1, CCK8 and apoptosis assays of HFSCs cotreated with siRNA-LEF1 and DPC-Exos were designed. Compared to the siRNA-LEF1 + PBS, HFSCs cotreated with siRNA-LEF1 and DPC-Exos increased cell proliferation viability (Figure 6A). Meanwhile, the apoptosis assay also showed that the addition of DPC-Exos could decrease the apoptosis rate in HFSCs after transfecting siRNA-LEF1 (Figure 6B).

## 3. Discussion

Hair follicles are complex and unique organs and form part of mammalian growth and development. The biogenesis and cyclical regeneration of HFs involve cellular communication between HFSCs and DPCs. Many pieces of research have shown that these two cell populations can influence each other. For example, HSFCs can secrete AIMP1-derived peptides that stimulate DPCs and promote hair growth [41], while DPCs can release growth factors that promote HFSC differentiation and proliferation [42]. Because of their ability to stimulate HFSCs, DPCs are master regulators of HF cycling [4]. Therefore, communication between HFSCs and DPCs is essential for HF growth and development.

Cell-to-cell communication is a cornerstone in the modulation of cellular activity, and exosomes, with their ability to deliver intercellular signals, are central to this [43,44]. As a signal carrier, exosomes have been shown to function in many systems, ranging from animal, in which rat MSC-derived exosomes promote chondrocyte proliferation [45], to man, where seminal fluid extracellular vesicles promote in vitro decidualization of human endometrial stromal cells [46]. Regarding hair follicle development, DPC-Exos have been shown to deliver miR-218-5p, stimulating β-catenin signaling and enhancing hair regeneration [47]. In human hair follicle cultures, DPC-derived exosomes have also promoted HFSC differentiation and enhanced hair growth [48]. This study provided additional evidence using Ki67 staining, CCK8 viability assay, and annexin staining of apoptotic cells to demonstrate that DPC-Exos can promote HFSC proliferation and inhibit HFSC apoptosis, consistent with previous research [27,28].

Our transcriptomic analysis of DPC-Exos-treated HFSCs identified 3702 DEGs, suggesting that DPC-Exos have the potential to regulate many aspects of HFSC function, such as proliferation and differentiation. Previous studies of dermal papilla cells showed that expression of signaling molecules, such as *WNT3*, *WNT4*, *WNT10B*, *AXIN2*, and *LEF1*, are enhanced during the anagen phase of the hair cycle [49]. In human skin, LEF1 guides the spacial organization of hair follicles and regulates the fate of epithelial cells [50]. In mice, the LEF/TCF protein complex functions alongside the Wnt-β-catenin pathway to regulate many genes involved in HF development [51]. In the skin of the Gansu Alpine Merino, LEF1 may facilitate the maturation of secondary HFs [52]. These pieces of research illustrate the impact of LEF1 on HF growth and development. Here, it was also found that DPC-Exo treatment significantly increases the expression of LEF1 mRNA in HFSCs. Overexpression of LEF1 can upregulate hair growth-related genes, promote HFSC proliferation, and inhibit HFSC apoptosis, while knockdown of LEF1 reverses these effects. These results suggest that DPC-Exos may regulate HFSC function by stimulating LEF1 expression within these cells.

In conclusion, DPC-Exos plays a crucial role in promoting HFSC proliferation. A total of 3702 genes were identified that were differentially expressed in HFSCs exposed to DPC-Exos. The key gene selected from this screening, *LEF1*, can promote cell proliferation and regulate HF growth and development. These findings provide novel insights into the function of intercellular communication in regulating HF growth and development.

## 4. Materials and Methods

### 4.1. Experimental Animal

All animal experiments were approved by the Yangzhou University Animal Care and Use Committee (approval number: 202103358). For DPC and HFSC isolation, dorsal skins of 6-month-old Angora rabbits were harvested under Zoteil-50 anesthesia. Wounds were cleaned with iodine solution after surgery, and the animals were allowed to recover under controlled housing conditions (12 h light/dark cycle, free access to water/food).

### 4.2. Cell Isolation and Culture

DPCs and HFSCs were isolated as previously described [53]. DPCs were cultured in Mesenchymal Stem Cell Medium (ScienCell, SanDiego, CA, USA). HFSCs were cultured in DMEM/F12 (Gibco, Grand Island, NY, USA), 2% Foetal Bovine Serum (SERANA, Pessin, Germany), 10 ng/mL EGF, 10 ng/mL insulin, and 0.4μg/mL hydrocortisone. DPC-Exos were collected from passage 3 DPCs. Passage 3 HFSCs were used for transcriptomic analysis.

### 4.3. Exosome Extraction and Staining

Exosomes were isolated from the DPC culture medium using the Total Exosome Isolation Kit (Lot: 4478359, Invitrogen, Carlsbad, CA, USA). Briefly, DPC supernatant was centrifuged to remove cell debris, followed by concentration with Ultra-15 centrifugal filter units (100-kDa cutoff, Millipore, Boston, MA, USA) [47,54]. Exosome isolation reagent was added to the concentrated medium and incubated overnight, followed by ultracentrifugation at 10,000× *g* for exosome harvesting. For DiI labeling, 10 mg of DiI was added to the culture medium for 10 min. The staining was terminated by adding 10 mL PBS, and the exosomes were extracted as above.

### 4.4. Transmission Electron Microscopy

A 20 μL sample of diluted exosomes was dropped onto a copper mesh, fixed with uranyl acetate solution, and air-dried at room temperature. Exosomes were imaged with transmission electron microscopy (Hitachi HT 7800, Tokyo, Japan) at 80 kV.

### 4.5. Nanoparticle Tracking Analysis (NTA)

Exosomes were diluted in PBS and analyzed on a Flow NanoAnalyzer instrument (NanoFCM, Nottingham, UK) to detect particle size and concentration of exosomes.

### 4.6. Immunofluorescence Staining

HFSCs were washed with PBS, fixed with 4% paraformaldehyde, and incubated with Ki67 primary antibody (1:250, Cat No. 27309-1-AP, Proteintech, Wuhan, China) overnight. Detection was performed with an antirabbit secondary antibody (1:20,000, Cat No.SA00009-2, Proteintech, China). Cell nuclei were stained with DAPI. Pictures were quantified with ImageJ software.

### 4.7. RNA-seq

RNA libraries were prepared and sequenced on the Illumina HiSeq platform to obtain 43–53 million reads per sample. Linker and low-quality reads were removed, and the clean reads were obtained by statistical filtering (Appendix A). The filtered reads were compared with the rabbit (*Oryctolagus cuniculus*) genome from Ensemble OryCun2.0 (Appendix A). All reads were submitted to the National Center for Biotechnology Information’s (NCBI) Short Read Archive (SRA) under accession number PRJNA907926. Sequence data were analyzed with the GO enrichment factors and KEGG pathway to predict potential biological function.

### 4.8. RT-qPCR

Total RNA was extracted with the RNAsimple total RNA kit (Tiangen, Beijing, China), and the cDNA was synthesized with HiScript III RT SuperMix for qPCR (Vazyme, Nanjing, China) as instructed. RT-qPCR was performed with ChamQ SYBR qPCR Master Mix (Vazyme, Nanjing, China). The primers used are listed in Appendix A.

### 4.9. Automated Protein Analysis

Cellular protein levels were detected with the Wes^TM^ automatic protein expression analysis system (Protein Simple, San Jose, CA) [55]. Protein samples (1 μg/μL) were stained with the following antibodies: GAPDH (1:2000; Cat No. 60004-1-Ig), CD9 (1:100; Cat No. 20597-1-AP), TSG101 (1:100; Cat No. 67381-1-Ig), calnexin (1:100; Cat No. 66903-1-Ig), LEF1 (1:100; Cat No. 14972-1-AP), CCND1 (1:100; Cat No. 60186-1-Ig), HRP-conjugated Affinipure Goat Anti-Mouse IgG(H + L) (1:10,000; Cat No. SA00001-1), and HRP-conjugated Affinipure Goat Anti-Rabbit IgG(H + L) (1:10,000; Cat No. SA00001-2). All antibodies were from Proteintech, China.

### 4.10. Vector Construction and Cell Transfection

The mRNA sequence of rabbit LEF1 (GenBank accession no. XM_002717171.4) was used for the primers designed from the LEF1 CDS sequence, and the PCR was performed using Phanta Max Super-Fidelity DNA Polymerase (Vazyme) and subcloned into EcoR I and Xba I (QuickCut restriction enzyme, TaKaRa, Dalian, China) digested pcDNA 3.1(+) vector, which was subsequently called pcDNA3.1-LEF1. The small interfering RNA (siRNA)-LEF1 was designed and purchased from Shanghai GenePharma Co., Ltd. (Shanghai, China) The primers for the overexpression vector and siRNA sequence are listed in Appendix A. The plasmids were transfected by Lipofectamine 3000 (Invitrogen, Carlsbad, CA, USA), and the siRNAs were transfected by Lipofectamine 2000 (Invitrogen).

### 4.11. Cell Proliferation and Cell Apoptosis

Cells were replated into 96-well plates 8 hours after transfection to detect cell proliferation. CCK8 (Cell Counting Kit-8, Beyotime, Shanghai, China) reagent was added at 0, 24, 48, and 72 h and incubated for 2 h, and the absorbance was measured at 450 nm using Infinite 200 Pro (Tecan, Männedorf, Switzerland).

Cells were harvested two days after transfection, washed twice with precooled PBS, stained with FITC and PI (Annexin V-FITC/PI Apoptosis Detection Kit, vazyme, Nanjing, China), and detected by flow cytometry (CytoFLEX S, Beckman, Brea, CA, USA) within 1 h to detect apoptosis.

### 4.12. Data Analysis

All statistical analyses were performed using SPSS 22.0 (SPSS Inc., Chicago, IL, USA). Paired sample t-test was used to examine the OD value, apoptosis rate, relative gene expression, and protein expression. Each analysis comprises three or more biological replicates. All data are presented as mean ± SD.

## Figures and Tables

**Figure 1 ijms-24-03961-f001:**
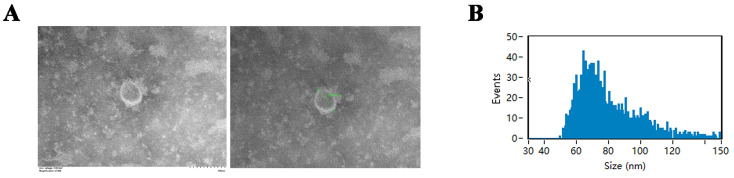
Characterization of DPC-Exos. (**A**) TEM images showing DPC-Exos shape and size (scale bars, 100 nm). (**B**) Nanoparticle tracking analysis (NTA) data showing DPC-Exos particle size and concentration distribution. (**C**) Wes^TM^ automatic protein expression analysis showing expression of DPC- and exosome-specific surface markers, calnexin, TSG101, and CD9, respectively. (**D**) Graphical representation of DiI-DPC-Exos treatment of HDSCs (left); brightfield microscopy image of passage 3 HFSCs (center); florescent microscopy image of HFSCs showing intracellular staining of DiI-DPC-Exos in red (right); scale bars, 50 μm.

**Figure 2 ijms-24-03961-f002:**
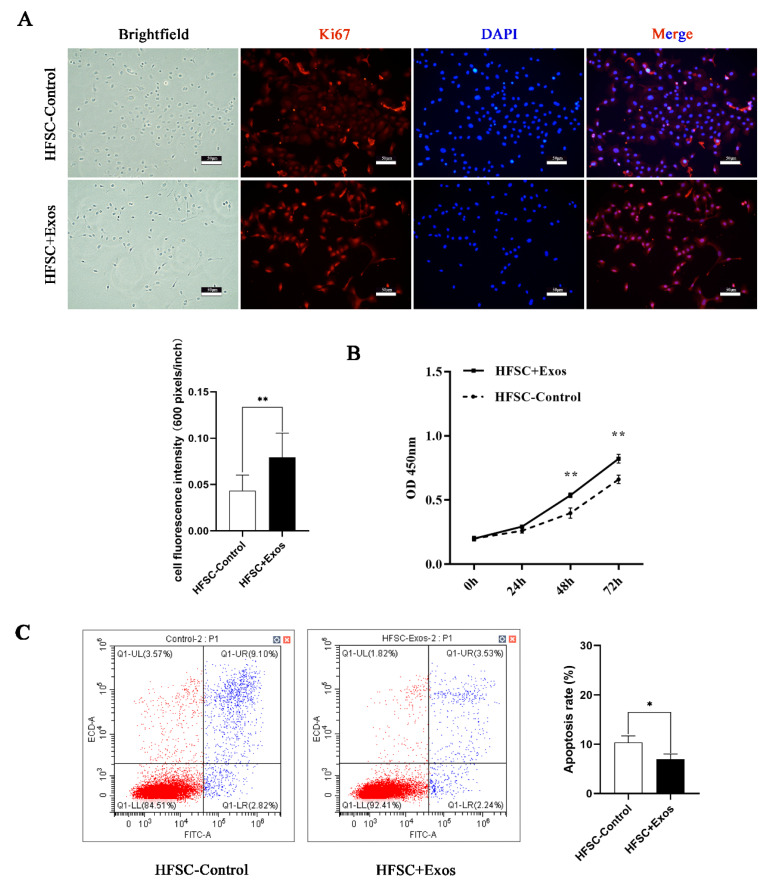
DPC-Exos promote proliferation and inhibit apoptosis of HFSCs. (**A**) Microscopic images of HFSCs stained with anti-Ki67 antibody (red) and the DAPI nuclear dye (blue). The cell fluorescence intensity was calculated. (scale bars, 50 µm). (**B**) The optical density reading of the CCK8 assay shows changes in HFSC viability after coculturing with DPC-Exos for 0, 24, 48, and 72 h. (**C**) Representative flow cytometry plots showing annexin-FITC stained HFSCs with and without DPC-Exos at 48 h after coculture (left); bar graph summarizing the flow cytometry data showing changes in apoptosis rate with DPC-Exos treatment (right). Data presented as mean ±SD, * *p* < 0.05, ** *p* < 0.01.

**Figure 3 ijms-24-03961-f003:**
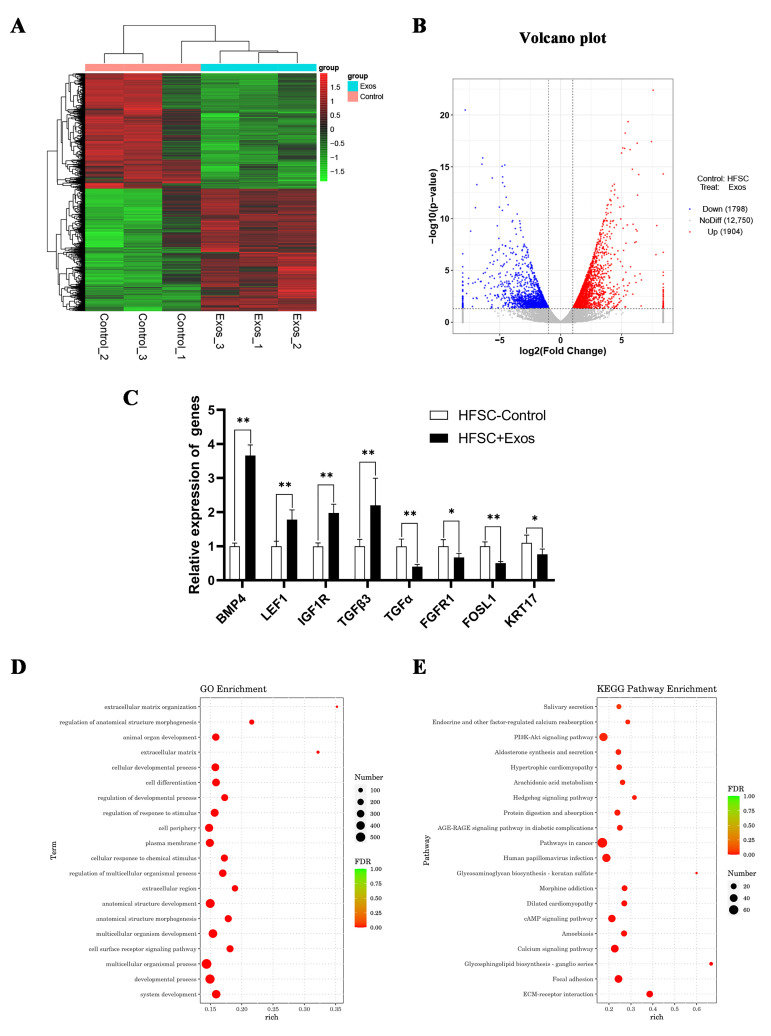
Effect of DPC-Exos treatment on HFSC gene expression. (**A**) Clustergram heatmap showing the levels of gene expression in HFSC-control and HFSC + Exos samples. (**B**) Volcano plot showing genes that are downregulated (blue), upregulated genes (red), or unchanged (gray) in HFSCs treated with DPC-Exos. (**C**) RT-qPCR data showing the relative expression of *BMP4*, *LEF1*, *IGF1R*, *TGFB3*, *TGFA*, *FGFR1*, *FOSL1*, and *KRT17* in HFSC-control and HFSC + Exos samples. (**D**) The upregulated GO enrichment-rich factors between HFSC-control and HFSC + Exos. (**E**) The upregulated KEGG enrichment pathways between HFSC-control and HFSC + Exos. Graph showing mean ±SD of gene expression normalized to control from triplicate experiments; * *p* < 0.05; ** *p* < 0.01.

**Figure 4 ijms-24-03961-f004:**
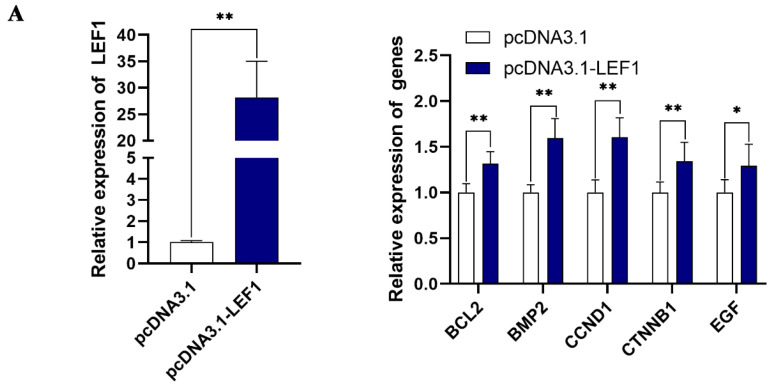
LEF1 regulates genes and proteins associated with hair follicle growth and development. (**A**) RT-qPCR data showing the relative expression of *LEF1* (left) and HF growth- and development-related genes (right) in HFSCs transfected with pcDNA3.1 or pcDNA3.1-LEF1. (**B**) RT-qPCR data showing the relative expression of *LEF1* in HFSCs transfected with siRNA control (siRNA-NC) or LEF1 knockdown siRNAs (siRNA-1915, -2088, and 2216; left); relative expression of HF growth- and development-related genes in HFSCs transfected with siRNA-1915 (renamed siRNA-LEF1; right). (**C**) Wes^TM^ automatic protein expression analysis showing LEF1 and CCND1 in HFSCs transfected with LEF1 overexpression or siRNA knockdown constructs (left); bar graph summarizing LEF1 and CCND1 protein expression with LEF1 overexpression and knockdown (right). All data presented as mean ± SD, * *p* < 0.05, ** *p* < 0.01.

**Figure 5 ijms-24-03961-f005:**
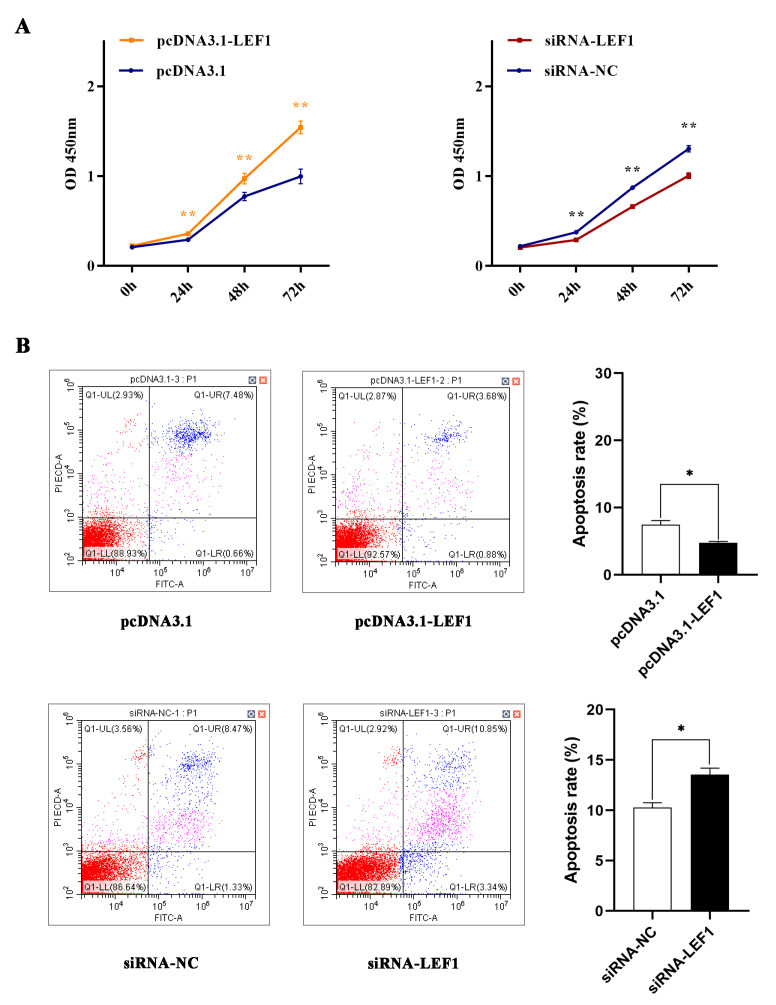
LEF1 stimulates proliferation and inhibits apoptosis of HFSCs. (**A**) The optical density reading of the CCK8 assay shows changes in the viability of HFSCs transfected with pcDNA3.1 or pcDNA3.1-LEF1 (left) and siRNA-NC or siRNA-LEF1 (right) after 0, 24, 48, and 72 h. (**B**) Representative flow cytometry plots showing annexin-FITC-stained HFSCs with and without LEF1 overexpression constructs (top row) and with and without LEF1 knockdown constructs (bottom row); bar graph summarizing the flow cytometry data showing changes in apoptosis rate with LEF1 overexpression and knockdown (right). Data presented as mean ± SD, * *p* < 0.05, ** *p* < 0.01.

**Figure 6 ijms-24-03961-f006:**
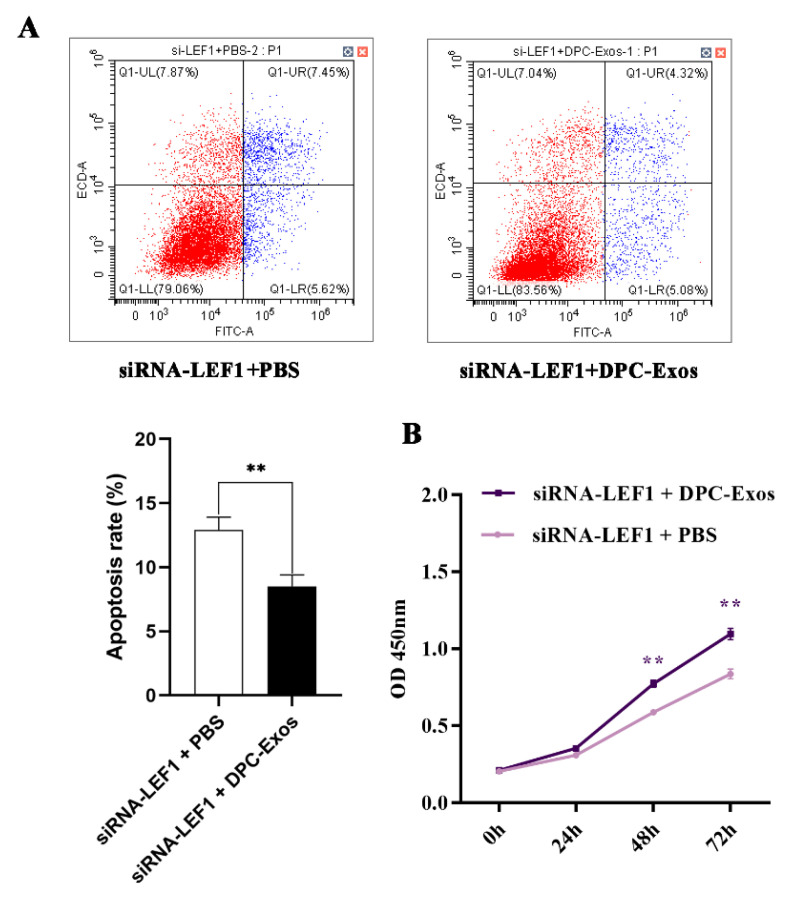
DPC-Exos could rescue the siRNA-LEF1 effect in HFSCs. (**A**) Representative flow cytometry plots showing annexin-FITC-stained HFSCs with siRNA-LEF1 and PBS or siRNA-LEF1 and DPC-Exos; bar graph summarizing the flow cytometry data showing changes in apoptosis rate with DPC-Exos in HFSCs after transfecting siRNA-LEF1. (**B**) The optical density reading of the CCK8 assay shows changes in the viability of HFSCs treated with siRNA-LEF1 and PBS or siRNA-LEF1 and DPC-Exos after 0, 24, 48, and 72 h. Data presented as mean ±SD, ** *p* < 0.01.

**Table 1 ijms-24-03961-t001:** Differentially expressed genes identified in DPC-Exos-treated HFSCs.

Gene Symbol	Gene ID	Log_2_ Fold Change (Exos/Control)	Up/Downregulated in the Exos Group
LAMC1	ENSOCUG00000013047	1.686131861	Up
BMP4	ENSOCUG00000011097	1.790001763	Up
LAMB2	ENSOCUG00000005101	1.556217384	Up
FGFR4	ENSOCUG00000003380	3.394558779	Up
LEF1	ENSOCUG00000010436	1.284592993	Up
Wnt6	ENSOCUG00000024007	2.221592693	Up
Wnt10b	ENSOCUG00000009580	1.73244649	Up
TGFβ3	ENSOCUG00000011415	2.904669796	Up
TGFβI	ENSOCUG00000004928	2.18361428	Up
IGF1R	ENSOCUG00000014795	1.227879353	Up
PDGFRL	ENSOCUG00000005211	2.312868441	Up
PDGFRB	ENSOCUG00000001725	2.156986092	Up
PDGFA	ENSOCUG00000005660	1.612847999	Up
IGFBP6	ENSOCUG00000001412	3.762560781	Up
IGFBP4	ENSOCUG00000002828	2.680551816	Up
TGFα	ENSOCUG00000029636	−2.701655579	Down
FGF5	ENSOCUG00000012418	−3.741909027	Down
KRT78	ENSOCUG00000029278	−3.493454975	Down
KRT80	ENSOCUG00000013920	−3.955347833	Down
KRT17	ENSOCUG00000005596	−1.175643763	Down
PIK3C2A	ENSOCUG00000005541	−3.059413655	Down
SLC5A1	ENSOCUG00000017569	−4.138033917	Down
ATG9B	ENSOCUG00000008673	−3.663322856	Down
EGR3	ENSOCUG00000000194	−1.099231618	Down
MAP3K9	ENSOCUG00000003204	−1.018089075	Down
ACTA1	ENSOCUG00000007566	−2.51892745	Down
FOSL1	ENSOCUG00000011143	−1.0225563	Down
ADGRF4	ENSOCUG00000013020	−1.50194114	Down
USP9X	ENSOCUG00000007198	−1.978925923	Down
USP33	ENSOCUG00000006721	−1.784859106	Down
ATAD5	ENSOCUG00000015229	−2.599828257	Down

## Data Availability

All data supporting our findings are included in the manuscript.

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
