# Peer review of "Dermal PapillaCell-Derived Exosomes Regulate Hair Follicle Stem Cell Proliferation via LEF1"

_ijms, 2023, doi:10.3390/ijms24043961_

Round 1

Reviewer 1 Report

Manuscript ID: IJMS-2164188
Type of manuscript: Article
Title: Dermal papilla cell-derived exosomes regulate hair follicle stem cell proliferation via LEF1
Authors: Jiali Li, Bohao Zhao, Shuyu Yao, Yingying Dai, Xiyu Zhang, Naisu Yang, Zhiyuan Bao, Jiawei Cai, Yang Chen, Xinsheng Wu *
Submitted to section: Molecular Biology

This manuscript tried to demonstrate that the cellular events (proliferation and apoptosis) of hair follicle stem cells (HFSCs) were regulated by exosomes from dermal papilla cells (DPCs) during the cyclic growth of hair follicles. The authors investigated the molecular basis via RNA seq and further validated the role of LEF1 in HFSC proliferation and apoptosis. However, Zhou et al. (reference #27) and Yan et al. (reference #28) presented quite similar results as this manuscript; therefore, the results in this manuscript seem not novel. In addition, the interpretation, interpretation, and presentation of the results should be improved.

Major concerns:

1. The authors should clearly state in the introduction the main difference between this work and Yan H et al. (reference 28). As Yan H et al. showed that micro RNAs from the exosome of DPCs regulate the proliferation of HFSCs via LEF1, the novelty of the work in this manuscript could be significantly lowered.

2. If the authors speculate that DPC-exo increased LEF1 expression, thereby stimulating the HFSC proliferation, an experiment that shows the DPC-exo rescue siLEF1 effect in HFSC will be needed. It will demonstrate the dependency of LEF1 expression upon DPC-exo-mediated regulation of HFSC cellular proliferation and apoptosis.

3. The description for control treatment against the DPC-derived exosome treatment in HFSCs was missing. The authors should include what they treated in control in HFSCs.

4. The improvement of data presentation

- Figure 2A: It will be helpful to show the quantitative data for immunocytochemistry images.

- Figure 3C: Any downregulated genes in DPC-exosome-treated HFSCs should be included.

Minor concerns:

1. There are several typos and grammatical errors throughout the manuscript. They should be corrected.

Author Response

Response to reviewers

Dear Editors and Reviewers:

Thank you for your letter and for the reviewers’ comments concerning our manuscript entitled “Dermal papilla cell-derived exosomes regulate hair follicle stem cell proliferation via LEF1” (ID: IJMS-2164188). Those comments are all valuable and very helpful for revising and improving our paper, as well as the important guiding significance to our research. We have studied the comments carefully and have made corrections. Revised portions are marked in red in the paper. The main corrections in the paper and the responses to the reviewer’s comments are as follows:

Major concerns:

  1. The authors should clearly state in the introduction the main difference between this work and Yan H et al. (reference 28). As Yan H et al. showed that micro RNAs from the exosome of DPCs regulate the proliferation of HFSCs via LEF1, the novelty of the work in this manuscript could be significantly lowered.

Response: Thanks for reviewer’s suggestion, we have added this part according to the Reviewer’s suggestion.

  1. If the authors speculate that DPC-exo increased LEF1 expression, thereby stimulating the HFSC proliferation, an experiment that shows the DPC-exo rescue siLEF1 effect in HFSC will be needed. It will demonstrate the dependency of LEF1 expression upon DPC-exo-mediated regulation of HFSC cellular proliferation and apoptosis.

Response: Thanks for reviewer’s suggestion, we have added this part to Figure 6.

  1. The description for control treatment against the DPC-derived exosome treatment in HFSCs was missing. The authors should include what they treated in control in HFSCs.

Response: Thanks for reviewer’s suggestion, we have added this part according to the Reviewer’s comments. HFSC-control group was added 100ul PBS for negative control.

  1. The improvement of data presentation

- Figure 2A: It will be helpful to show the quantitative data for immunocytochemistry images.

Response: Thanks for reviewer’s suggestion, we have added this part to Figure 2A.

- Figure 3C: Any downregulated genes in DPC-exosome-treated HFSCs should be included.

Response: Thanks for reviewer’s suggestion, TGFα, FGFR1, FOSL1, KRT17 were downregulated in DPC-exosome-treated HFSCs.

Minor concerns:

  1. There are several typos and grammatical errors throughout the manuscript. They should be corrected.

Response: Thanks for reviewer’s suggestion, we have made correction according to the Reviewer’s comments.

We tried our best to improve the manuscript and made some changes in the manuscript. These changes will not influence the content and framework of the paper. And here we did not list the changes but marked in red in revised paper.

We would like to thank the referee again for taking the time to review our manuscript.

Yours sincerely,

Jiali Li

Reviewer 2 Report

The paper deals with the exosomes analysis of dermal papilla cells to HFSCs. The topic is hot. The paper is well made and brings new knowledge.

 Specific comments 

 1.In the process of submission, it is best to add line number, convenient for reviewers to review the paper.

2.Why there is no internal reference gene in Fig.1C and why WB is not used

3.The immunofluorescence results of Fig.2A  should be quantified.

4.Add “.”between Fig2 heading and (A).

5.RT-qPCR means Real-Time Quantitative PCR? What is the meaning of QPCR? fig3C uses QPCR, and the full text should be unified.

6.Why was lef1 selected for fuction validation instead of the gene with the greatest difference in expression?

7.Table 1 shows the differentially expressed top15 gene? Information about other differentially expressed genes should also be presented in a supplementary table. 

8.We further confirmed at the protein level that CCN1 level is dependent on LEF1 expression (Fig.3C).CCN1 should be “CCND1” ?

Author Response

Response to reviewers

Dear Editors and Reviewers:

Thank you for your letter and for the reviewers’ comments concerning our manuscript entitled “Dermal papilla cell-derived exosomes regulate hair follicle stem cell proliferation via LEF1” (ID: IJMS-2164188). Those comments are all valuable and very helpful for revising and improving our paper, as well as the important guiding significance to our researches. We have studied the comments carefully and have made corrections. Revised portions are marked in red in the paper. The main corrections in the paper and the responses to the reviewer’s comments are as follows:

1.In the process of submission, it is best to add line number, convenient for reviewers to review the paper.

Response: Thanks for reviewer’s suggestion, we have made correction according to the Reviewer’s comments.

2.Why there is no internal reference gene in Fig.1C and why WB is not used

Response: We are very sorry for our negligence of the internal reference gene, we have made correction according to the Reviewer’s comments. The traditional WB method requires a large amount of samples. All HFSCs in this experiment can only be isolated and used immediately, and only 5 generations of HFSCs can be used for each isolation. The WesTM automatic protein expression analysis system only needs no more than 5ul of sample volume per well. In addition, the traditional WB method takes almost two days, but the fully automatic Western blot analyzer only takes a few hours to obtain the experimental data. All in all, this is why we chose Wes.

3.The immunofluorescence results of Fig.2A should be quantified.

Response: Thanks for reviewer’s suggestion, we have made correction according to the Reviewer’s comments.

4.Add “.”between Fig2 heading and (A).

Response: Thanks for reviewer’s suggestion, we have made correction according to the Reviewer’s comments.

5.RT-qPCR means Real-Time Quantitative PCR? What is the meaning of QPCR? fig3C uses QPCR, and the full text should be unified.

Response:Thanks for reviewer’s suggestion, we are very sorry for our negligence of the abbreviation, we have made correction according to the Reviewer’s comments.

6.Why was lef1 selected for fuction validation instead of the gene with the greatest difference in expression?

Response: Thanks for reviewer’s suggestion, LEF1 has been reported many times that it is related to hair follicle development, and our team previous research also found LEF1 has a positive regulatory effect on hair follicles. The greatest difference in expression gene (IFI44), there are few studies on hair follicles.

7.Table 1 shows the differentially expressed top15 gene? Information about other differentially expressed genes should also be presented in a supplementary table.

Response: Thanks for reviewer’s suggestion, Table 1 just summarizes the DEGs that may be involved in HF growth and development. We have added all the differentially expressed genes to supplementary table S5.

8.“We further confirmed at the protein level that CCN1 level is dependent on LEF1 expression (Fig.3C).” CCN1 should be “CCND1” ?

Response: Thanks for reviewer’s suggestion, we have made correction according to the Reviewer’s comments.

We tried our best to improve the manuscript and made some changes in the manuscript. These changes will not influence the content and framework of the paper. And here we did not list the changes but marked in red in the revised paper.

We would like to thank the referee again for taking the time to review our manuscript.

Yours sincerely,

Jiali Li

Round 2

Reviewer 1 Report

The major concerns that I raised were answered.